psychology

self-correction, metascience, publication practices

**Author for correspondence:**
Tom Heyman
e-mail: t.d.p.heyman@fsw.leidenuniv.nl

# Correction notices in psychology: impactful or inconsequential?

Tom Heyman[1,2,†,‡] and Anne-Sofie Maerten[1,‡]

[1]KU Leuven, Leuven, Belgium
[2]Leiden University, Leiden, Netherlands

  TH, 0000-0003-0565-441X; A-SM, 0000-0003-2710-0936

Science is self-correcting, or so the adage goes, but to what extent is that indeed the case? Answering this question requires careful consideration of the various approaches to achieve the collective goal of self-correction. One of the most straightforward mechanisms is individual self-correction: researchers rectifying their *own* mistakes by publishing a correction notice. Although it offers an efficient route to correcting the scientific record, it has received little to no attention from a metascientific point of view. We aim to fill this void by analysing the content of correction notices published from 2010 until 2018 in the three psychology journals featuring the highest number of corrections over that timespan based on the Scopus database (i.e. *Psychological Science* with $N = 58$, *Frontiers in Psychology* with $N = 99$ and *Journal of Affective Disorders* with $N = 57$). More concretely, we examined which aspects of the original papers were affected (e.g. hypotheses, data-analyses, metadata such as author order, affiliations, funding information etc.) as well as the perceived implications for the papers' main findings. Our exploratory analyses showed that many corrections involved inconsequential errors. Furthermore, authors rarely revised their conclusions, even though several corrections concerned changes to the results. We conclude with a discussion of current policies, and suggest ways to improve upon the present situation by (i) preventing mistakes, and (ii) transparently rectifying those mistakes that do find their way into the literature.

## 1. Introduction

As researchers, we devote our careers to uncovering interesting facts, developing theories, improving research practices or advancing statistical procedures. Often, the results are published

[†]Methodology and Statistics Unit, Institute of Psychology, Leiden University, Wassenaarseweg 52, 2333 AK Leiden, The Netherlands.
[‡]Joint first authors.

in journals, books and conference proceedings. Nobody is without flaw, though and mistakes could infiltrate our work. Despite peer-review and other error-detection procedures, some of these mistakes find their way into the literature. Thus, rectifying them is critical to build an unbiased knowledge base. After all, science is assumed to be self-correcting.

There is more than one way to Rome, though. Correcting the scientific record can, for instance, be achieved via 'normal progression of a discipline' [1, p. 3]. By conducting research, spurious findings and flawed theories get exchanged for new ones. When it comes to correcting or updating empirical results, independent replication studies, which are gradually becoming more mainstream [2], may be considered the gold standard. However, replications often require substantial investments (e.g. time, money,…), or might even be completely unfeasible. Not to mention, replication studies can be flawed in their own right.

Fortunately, some erroneous conclusions can be corrected in a more efficient fashion. One could, for instance, try to analytically reproduce the reported results using the original data (if these are shared; e.g. [3]). Alternatively, one might use tools such as statcheck [4] to uncover mistakes in (the reporting of) statistical analyses, without needing the raw data. Yet another possibility is to assess the robustness of a finding by not merely following one particular data-analytic pathway, but many (a multiverse analysis; [5]).

That being said, there are legitimate concerns about the pace and efficiency of these self-corrective mechanisms. Moreover, they can even result in an unproductive back-and-forth, as illustrated by the oft-dismissive responses to criticism documented by Goldacre *et al.* [6].

Yet another approach to achieve the collective goal of scientific self-correction is via *individual self-correction*. Whenever authors become aware of an error in one of their publications, they could set the record straight by issuing a correction notice. Such statements often contain information on which aspects of the work requires correction, as well as their impact on the conclusions. When the error is considered too egregious, or fraudulent practices were discovered, the publication might get retracted altogether, though. Corrections and retractions are a fundamentally different form of scientific self-correction as it does not follow the typical progression of research [1]. Rather it involves erasing certain information from the scientific record and, in the case of a correction, replacing it with accurate information.

Of the various forms of scientific self-correction, retractions, 'failed' replications and reproducibility problems have received a great deal of attention from the research community, as well as the public in general. By contrast, correction notices remain largely unexplored [7], even though it seems a straightforward mechanism to rid the literature of certain mistakes.[1] Moreover, the mere publication of correction notices is taken to mean that (psychological) science is self-correcting. For instance, Grcar [8] used the number of corrections (and commentaries) divided by the number of published articles to examine how self-correcting the written scientific record is.

So, even though correction notices are recognized as a form of scientific self-correction [1,8,9], one might wonder how much they truly impact the underlying knowledge base. Do they also yield new insights, and revise theories like other forms of scientific self-correction, or do they mostly involve superficial aspects of the scientific record?

To address this question, we analysed the content of correction notices published in three psychology journals. More specifically, we examined what aspect of the original paper required correction: the hypotheses, methodology, data-handling, data-analyses, reporting, references, aesthetics, metadata (e.g. author order, affiliations, funding information etc.), or something else. Furthermore, we assessed the perceived impact on the paper's initial conclusions, both from the authors' point of view and that of a reader, thus gauging whether a certain correction substantially affected the literature.

## 2. Method

Following Grcar [8], we used Scopus to search for correction notices, as it has a designated document type for such publications (i.e. document type=er, short for erratum). Note that the term erratum is sometimes reserved for corrections following mistakes by the publisher (e.g. mishaps in the copyediting process). However, in Scopus, it also encompasses corrections issued by the author(s) as well as retractions.

Considering publications in psychological research, it appears that the corrections-to-articles ratio remained relatively stable from 2003 until 2011, after which it seems to steadily increase until 2017

---

[1]This is not a critique on replication studies or any other form of scientific self-correction. Individual self-correction is no silver bullet, but it might be more efficient at times. For example, if authors find a critical mistake in their analyses, submitting a transparent correction notice might be preferred over having an entire team of independent researchers engage in a large-scale replication study.

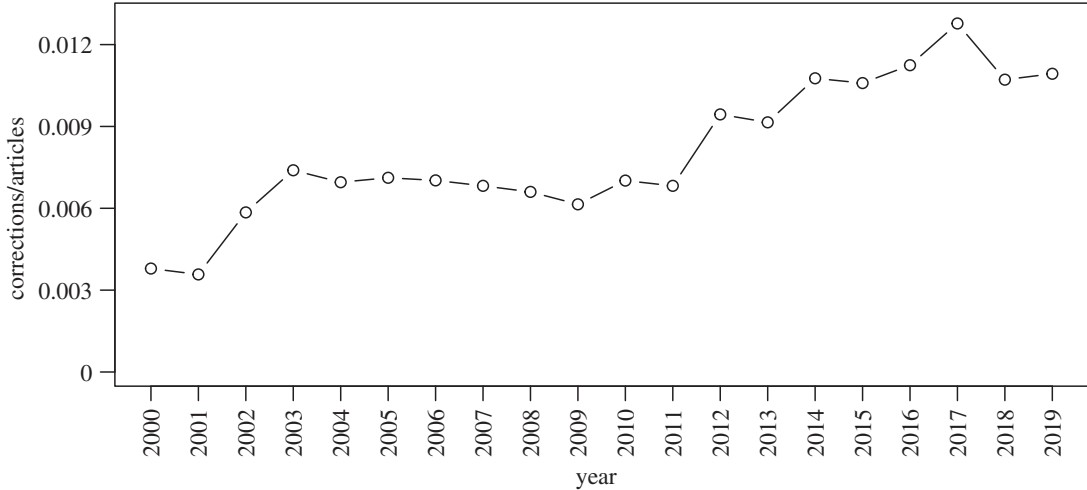

**Figure 1.** Evolution over time of the number of corrections divided by the number of articles published in psychology. The time period is restricted for presentation purposes. The numbers were extracted from Scopus on 28 January 2020.

(figure 1). In this paper, we focused on recent correction notices, such that relatively novel developments (e.g. increased access to data) could potentially be captured, but we also included those from the years immediately before the apparent (temporary) uptick in prevalence. Thus, we ultimately considered corrections published from 2010 until 2018.[2]

This selection resulted in 4495 documents, which we opted to further narrow down in order to make it practically feasible. That is, we selected the three journals with the highest number of corrections published in the aforementioned time period, namely *Frontiers in Psychology* (99 corrections, relative to 9681 articles published in that timeframe), *Journal of Affective Disorders* (57 corrections, 5113 articles) and *Psychological Science* (58 corrections, 1970 articles). Even though this is a non-random sample, it does cover different publishers (i.e. Frontiers, Elsevier and SAGE), publication models (i.e. online, pay-to-publish versus traditional, pay-to-read) and disciplines within psychology. It also follows other meta-scientific studies in psychology that have opted to focus on multiple articles from a small number of journals (typically one to four) rather than randomly selecting articles across an entire domain, thus keeping policies and quality standards constant (e.g. [3,10–13]).

Figure 2 depicts the corrections-to-articles ratio for the selected journals in the timeframe under consideration. The trends are comparable across the three journals in the sense that correction notices seem to have become somewhat prevalent only in recent years, which also supports the decision to focus on the period 2010–2018 rather than going back further. Note also that *Frontiers in Psychology* was only established in 2010. Hence, a certain 'correction lag' is to be expected, yet the evolution over time is similar to the other two journals. These figures should be interpreted with caution, though, as sample sizes are relatively small. In addition, they are based on the Scopus database, which, for instance, includes some duplicates.

Each correction was coded by both authors independently, using a prespecified scheme featuring four main variables: (i) which aspect(s) of the article required a correction, (ii) whether (some of) the results presented in the original paper changed after correction, (iii) whether the conclusion regarding the main findings remained unchanged according to the authors, and (iv) who initially discovered the error(s) in the original article (see table 1 for an overview of all categories, and https://osf.io/uyqra/ for more details including examples). More specifically, the corrected findings were first coded according to which part of conducting scientific research they pertained to. Note that a correction could concern more than one aspect (e.g. both the metadata and data-analysis of a given paper could contain an error). In such cases, we coded all aspects that were affected.

When the results presented in the correction notice differed from those of the original article, their pertinence was determined. Pertinent results were defined as those that have implications for the main findings and conclusions of the original paper. This is arguably a subjective judgement, and because many papers were beyond our area of expertise, it is possible that we misclassified some cases (we revisit this issue in the Discussion). Changed results that were deemed pertinent were

---

[2]We finished the coding of corrections in July 2019, which is why the selected time frame ends in 2018, unlike the *x*-axis of figure 1), which runs through 2019.

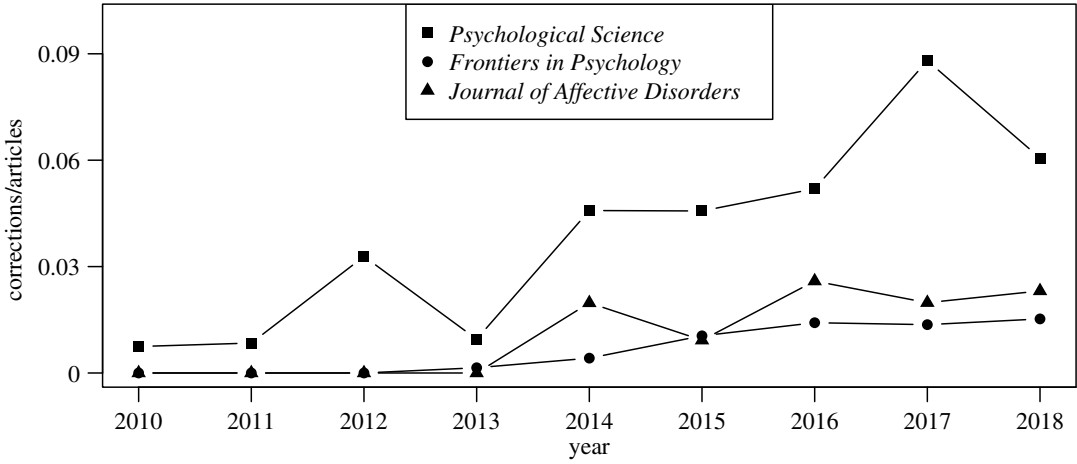

**Figure 2.** Evolution of the number of corrections divided by the number of articles published in *Psychological Science* (squares), *Frontiers in Psychology* (dots) and *Journal of Affective Disorders* (triangles) across the selected timeframe. The numbers were extracted from Scopus on 30 July 2020.

**Table 1.** Overview of the coding scheme.

| correction type | results[a] | | implications | discovered by |
|---|---|---|---|---|
| | result type | bolster conclusion | | |
| aesthetics | Bayes factor | no | different conclusion, different results | author(s) |
| data-analysis | effect size | yes | same conclusion, different results | journal |
| data-handling | *p*-value | unclear | same conclusion, same results | someone else |
| hypotheses | other | | unclear conclusion, different results | unknown |
| metadata | | | unclear conclusion, same results | |
| methodology | | | | |
| referencing | | | | |
| reporting | | | | |
| other | | | | |

[a]Only coded when the correction impacted one or more pertinent results.

further subdivided into four categories: *p*-values, standardized effect sizes, Bayes factors and a rest category including raw effect sizes (e.g. means, unstandardized regression coefficients, and the like; these were labelled as 'other' in table 1). For each pertinent change, the original value was compared with the corrected value, to assess whether the corrected value bolstered the original conclusion regarding that finding (e.g. values more extreme in the direction of the drawn conclusion versus values being less extreme or going in the opposite direction of what was claimed initially).[3]

Moreover, we also examined the conclusion of the correction notice. That is, we determined whether the authors explicitly state that their original conclusion remains unchanged, or requires revision. If they made no statement regarding the conclusion, or if the notice was deemed ambiguous in this regard, we classified it as unclear.

Lastly, for each correction, we examined who discovered the error in the original paper: the authors themselves, the journal it was published in, someone else, or whether the information provided in the correction notice was insufficient to deduce who discovered the error.

After both authors evaluated the correction notices, their results were compared. In case of disagreement, the correction in question was re-examined, and the most appropriate coding was selected after discussion between both authors. Retractions were filtered out, and thus not included in the following analyses.

---

[3]One could view this as a rather crude measure to evaluate whether a conclusion was bolstered, but theories or hypotheses tested within psychology are often somewhat vague in that they rarely specify the size of an effect *a priori*.

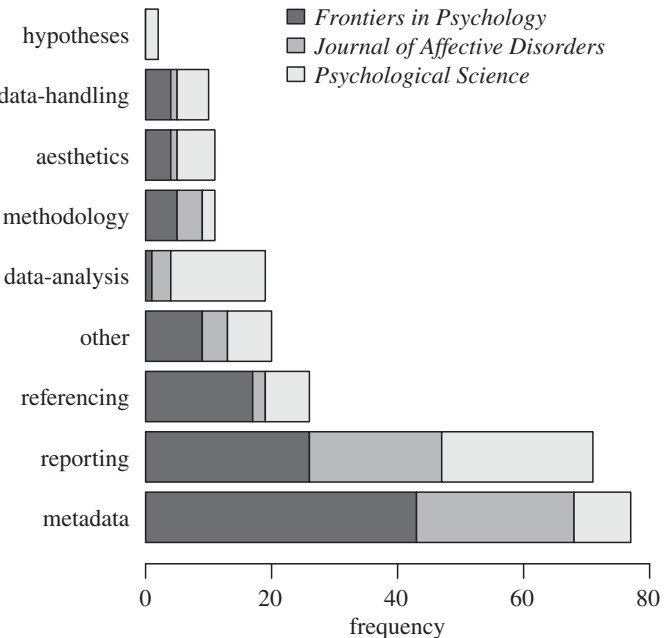

**Figure 3.** Distribution of correction types by journal.

Note that the current study did not seek to test any *a priori* hypotheses. Instead, it was exploratory in nature, as little is known about this domain. Also, the study's protocol was not formally pre-registered, even though the coding scheme was designed in advance. As such, the results should be interpreted with due caution.

## 3. Results

As illustrated by figure 3, corrections mostly involved reporting errors ($N = 71$, 28.74%), incorrect referencing ($N = 26$, 10.53%) or metadata ($N = 77$, 31.17%). Errors that typically impact the originally presented results and conclusions, such as those concerning the hypotheses, methodology, data-handling and data-analysis, comprised a smaller percentage of corrected errors (i.e. $N = 42$, 17.00%), whereas errors pertaining to aspects that typically do not substantially affect the scientific knowledge base such as aesthetics, referencing or metadata accounted for a larger percentage of errors (i.e. $N = 114$, 46.15%). Note that reporting errors could affect different aspects of a paper that may or may not impact its main findings, hence they were not included in this comparison. The same goes for corrections that were coded as 'other' because they did not fall in any of the eight categories (e.g. incorrect interpretation of the results or previous findings, adding new information in the form of appendices, etc.). Figure 3 also reveals that the distribution of correction types differs across journals to some extent. Most notably, *Psychological Science* published relatively more corrections notices mentioning mistakes in the data-analysis, whereas issues regarding metadata were less prevalent compared with the other two journals.[4]

Across all categories and journals, 65 corrections (30.37%) involved changes to some of the results. Out of those corrections that presented different results, 3 (4.62%) mentioned that the conclusion regarding their main findings changes. Twenty-nine (44.62%) did not explicitly address the impact on the original conclusions, whereas 33 (50.77%) suggested that their main conclusions still hold. A break-down by journal revealed that most corrections in *Frontiers in Psychology* claimed that the main conclusions remain the same (i.e. out of 16 corrections with different results 14 or 87.50% stated that the conclusions still hold). This percentage was much lower for the other two journals (i.e. 16 out of 30 or 53.33% for *Psychological Science*, and 3 out of 19 or 15.79% for *Journal of Affective Disorders*).

Often, the corrected results were not strikingly different from the original findings. As such, it seems justifiable to claim that the original conclusions remain valid. However, in some cases, it can be misleading to present the conclusions as unchanged. For example, a *p*-value that is no longer

[4]Note that a given correction can involve multiple aspects, but this rarely occurred. Of the 214 corrections, 191 (89.25%) pertained to only one aspect.

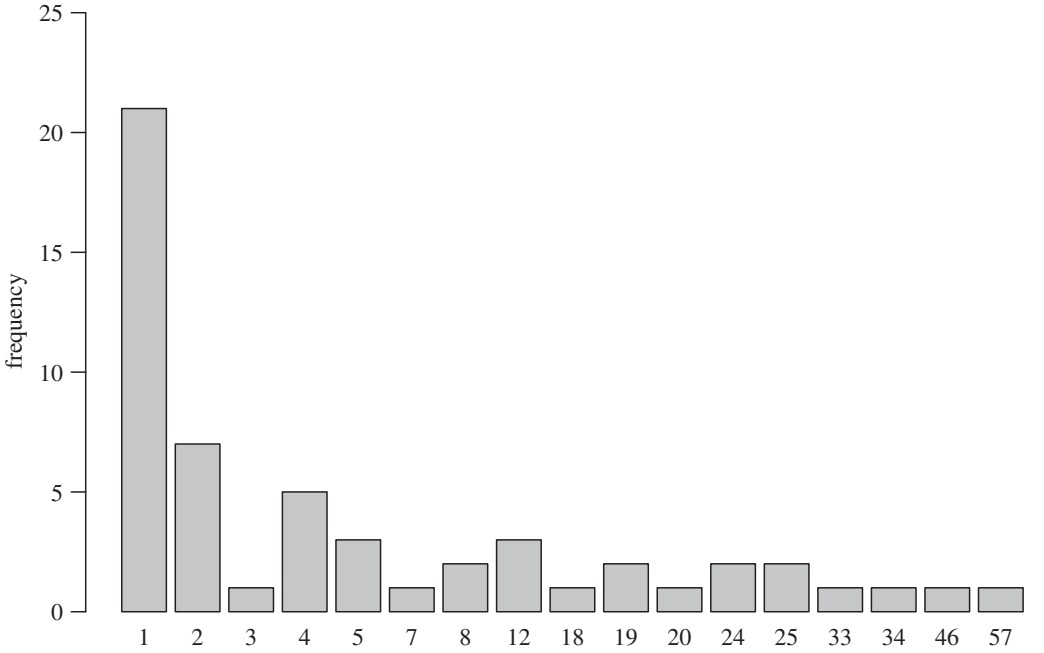

**Figure 4.** Distribution of number of pertinent results per correction.

considered statistically significant, or a 50% reduction in the effect size, could be viewed as substantial changes (depending on the statistical philosophy).

To examine this, we first assessed for each correction that presented changed results ($N = 65$), whether it involved a pertinent finding. Fifty-four (83.08%) of such corrections included at least one pertinent corrected result. Compared with the total number of corrections published in each respective journal, those appearing in *Psychological Science* often involved changes to pertinent results (i.e. 27 out of 58 corrections or 46.55%). This figure was much lower in the other two journals (i.e. 13 out of 99 corrections or 13.13% for *Frontiers in Psychology* and 14 out of 57 corrections or 24.56% for *Journal of Affective Disorders*).

Next, we compared the original values with the reported values and evaluated the impact on the conclusions. Of all the 476 pertinent results in total, 73 (15.34%) bolstered the original conclusion, whereas 151 (31.72%) weakened the original conclusion. In the remaining cases, it was unclear how the change affected the conclusions. This occurred fairly often, and for a variety of reasons such as inexact reporting of the results, inexperience of the coders with the particular domain, or inherent subjective decisions (e.g. smaller effect sizes that were estimated more precisely or vice versa). This distribution was fairly similar across journals with 6 out of 46 (13.04%), 60 out of 380 (15.79%) and 7 out of 50 (14.00%) pertinent results bolstering the original conclusion, versus 14 (30.43%), 123 (32.37%) and 14 (28.00%) pertinent results weakening the original conclusion for *Frontiers in Psychology*, *Psychological Science* and *Journal of Affective Disorders*, respectively. These results also show that the number of pertinent results was much higher for corrections in *Psychological Science* compared with the other two journals.

Comparing the amount of pertinent results with the number of corrections reveals that certain corrections feature many pertinent results. This is illustrated by figure 4, which depicts the distribution of the number of pertinent results per correction. Thus, pertinent results are not independent from one another as some come from the same article. Consequently, if one result weakens the conclusion of a particular article, others from the same correction might have a comparable effect. To take this dependency into account, we calculated a second measure of perceived impact, as judged by the coders, in which every study got an equal weight. The results were fairly similar: the prevalence of bolstering conclusions was 15.54% relative to 24.65% weakening the original conclusion (a break-down by journal also revealed comparable results).

Whether or not a result is judged to bolster the original conclusions is of course a subjective assessment to a certain extent, and should therefore be taken with a pinch of salt. Indeed, the coders' initial classification was often not exactly the same. Considering the corrections, the coders agreed on all aspects (i.e. the correction type, perceived implications and source of the correction) in 52.63% of the cases. When focusing on pertinent results, complete agreement was 38.03%. Both numbers, and the latter in particular, do get deflated because of missing evaluations (i.e. aspects or outcomes not considered by one of the coders), rather than actual disagreements.

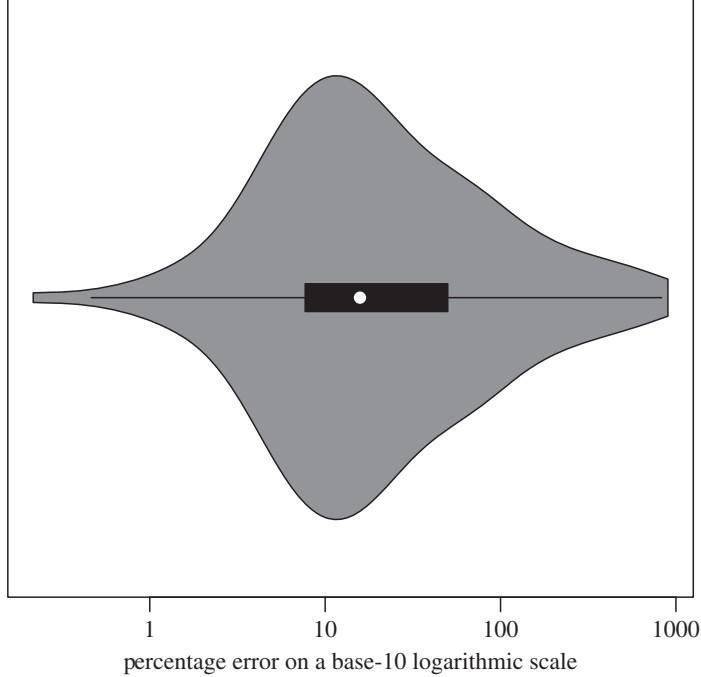

1            10            100            1000

percentage error on a base-10 logarithmic scale

**Figure 5.** Violin plot illustrating the percentage error distribution for the effect sizes.

Nevertheless, to address concerns about objectivity, we also compared the raw values, before and after correction, as these are not (or less) subject to alternative interpretations. To this end, we focused on the *p*-values and effect sizes that were exactly reported in both the original paper and the correction notice (none of the corrections mentioned Bayes factors). To compare different measures of effect size, we adopted the percentage error (PE) metric proposed by Hardwicke *et al.* [3] for evaluating reproducibility errors. It divides the absolute difference between the original and correct effect size by the absolute value of the original effect size:

$$PE = \frac{|correct - original|}{|original|}.$$

Hardwicke and colleagues considered PEs greater than 10 to be major numerical errors, though this cut-off is somewhat arbitrary. Figure 5 shows the distribution of PEs from the correction notices, plotted on a base-10 logarithmic scale for presentation purposes and collapsed across journals as the results were comparable. Interestingly, most corrected effect size would be considered major numerical errors, whereas only 5% of the discrepancies discovered in Hardwicke *et al.*'s analytic reproducibility project fell in this category.

Furthermore, we also examined the original and corrected (pertinent) *p*-values, though not in terms of PE as they are all on the same scale (figure 6). We again collapsed across journal, because most of these values came from just one journal (i.e. *Psychological Science*). One might argue that more originally significant *p*-values became statistically insignificant (using the typical alpha-level of 0.05) after correction, than vice versa. However, there are a number of caveats, which also apply to the above analysis of effect size. The sample size was somewhat small, and the underlying values were not independent from one another as some came from the same corrected article. In addition, publication bias might filter out some non-significant incorrect results that, upon correction, would have turned out statistically significant. Hence, one should interpret these figures with due caution.

Finally, most of the corrections provided no information regarding who discovered the error(s) in the original paper (i.e. 181 or 84.58%). Only, 13 (6.07%) mentioned that someone else found an error in the original paper, another 14 (6.54%) were issued by the journal they were published in and 6 (2.80%) came as a result of the authors discovering the mistake(s) themselves. A break-down by journal showed that it is in general unclear who discovered the error(s) with 85 (85.86%), 47 (81.03%) and 49 (85.96%) of the corrections in, respectively, *Frontiers in Psychology*, *Psychological Science* and *Journal of Affective Disorders* providing no information in that regard.

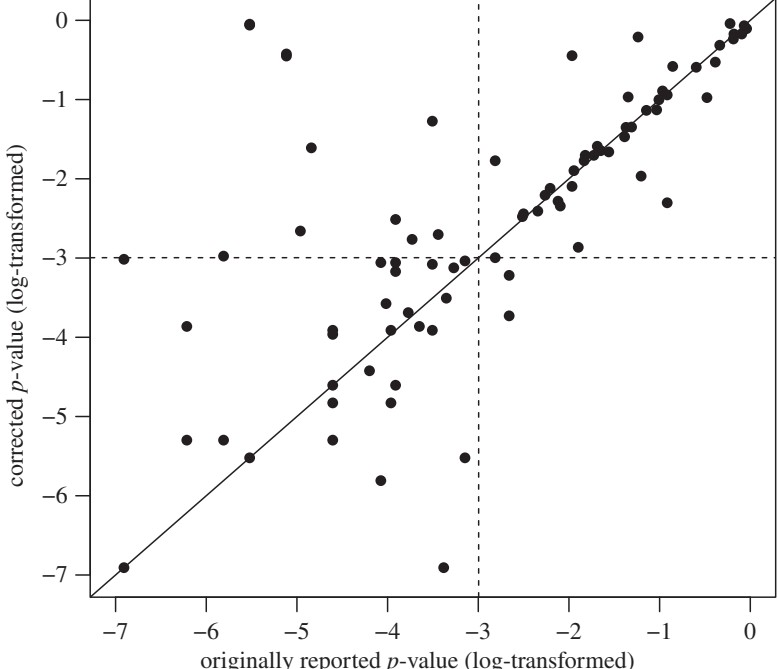

**Figure 6.** Scatter plot of original and corrected *p*-values. The dotted lines illustrate the commonly used 0.05 significance threshold.

## 4. Discussion

Based on an analysis of correction notices published between 2010 and 2018 in *Psychological Science*, *Frontiers in Psychology* and *Journal of Affective Disorders*, we can conclude that a substantial fraction only updates the scientific *record*, but not the underlying knowledge base as such. That is, a large number of notices concerned author order, the spelling of an author's name, their affiliation, omitted references, and the like. Even though these specific forms of individual self-correction all rectify a mistake of some kind, they do not revise flawed scientific notions. In that sense, the publication of correction notices is a poor indication of the degree to which (psychological) science is truly self-correcting, thus nuancing previous claims (e.g. [8]).

Naturally, a non-negligible number of notices did involve a correction of the study's outcome. However, there were very few instances where the authors explicitly indicated that this changed the conclusions of the original paper.[5] Most of the time, no explicit statement regarding the conclusion was provided. This could be because the authors considered the impact to be self-explanatory, because they were uncertain themselves, because they were hesitant to denounce their findings, etc. In some cases, it can be obvious that a conclusion remains unchanged, yet, in a substantial number of corrections without an explicit statement, their impact can be rather ambiguous. This is especially striking for corrections published in *Frontiers in Psychology*, because they have a template ('Author Guidelines', n.d. [14]) which by default includes the sentence: 'The authors apologize for this error and state that this does not change the scientific conclusions of the article in any way.' Consequently, the omission of this sentence might imply that the conclusions do *not* remain the same, though this is pure speculation. The policy is interesting in its own right, because it seemingly confirms or instils the notion that corrections are meant to reaffirm the status quo. Indeed, nearly all corrections involving changes to the results claimed that the conclusions remained the same, which was not the case for the other two journals under consideration. Instead of using such a default, one could ask authors to choose between different outcome options, thus engaging them in more critical reflection. Furthermore, creating a distinct category for corrections that concern authorship, affiliations, and the like, would obviate rather bizarre statements such as 'The name of author X was misspelled. The authors apologize for this error and state that this does not change the scientific conclusions of the article in any way.' Although it has the advantage of removing any ambiguity, using it like a one-size-fits-all statement also devalues its meaning in general.

---

[5]One could argue that major revisions of one's original conclusions might end up as retractions instead. To address that possibility, we *post hoc* examined the content of all 16 retraction notices in our sample. Only, five of those were due to honest mistakes. The remainder were issued because of (alleged) scientific misconduct.

Even though most correction notices stated that the conclusions remain the same despite changes in the results, a more nuanced picture emerged from the present study. Comparing the original with the corrected results revealed that the latter weakened the papers' conclusions more often than they strengthened it. Of course, it is possible that some of the smaller quantitative differences need not prompt a qualitative revision of the original conclusion, yet the apparent imbalance between conclusion-bolstering versus conclusion-weakening corrections is still remarkable. By definition, corrections occur at the very end of the entire publication process, and different factors could have contributed to this ostensible asymmetry. For example, results that provided less support for a paper's main conclusion might have been checked more rigorously by the authors. Therefore, corrected results that end up bolstering the original conclusion might be more rare. Another caveat is that the coding itself could be biased. The coders (i.e. both authors of this paper) did have different backgrounds, which might mitigate that concern to some degree, though it seems impossible to remove all subjectivity from this process. Finally, it remains to be seen whether these results generalize to other journals within psychology. An analysis by journal did reveal that the asymmetry between conclusion-bolstering versus conclusion-weakening corrections was fairly constant across the three outlets under consideration, though.

## 4.1. Where is the self-correction?

Figure 1 showed that there is about 1 correction per 100 articles published within psychology. It is plausible to assume, however, that many errors go *un*detected. If we only focus on the results section, we know from previous studies that analytical reproducibility is far from perfect. For instance, Hardwicke *et al.* [3] found that 24 out of 35 examined articles published with open data in the journal *Cognition* contained at least one value that could not be reproduced independently. Even after assistance of the original authors, results from 13 of those articles were still not reproducible. Unpublished work by Artner and colleagues examining papers in four different journals showed similar results, as did a recent study by Hardwicke *et al.* [15] involving articles appearing in *Psychological Science*. Thus, we can safely assume that many more corrections should be published, if all errors would get detected.

Furthermore, surveys like the one from John *et al.* [16] revealed a disconcertingly high percentage of researchers engaging in questionable research practices such as selectively reporting studies that 'worked', and adjusting data collection based on whether the results reached statistical significance. Now that the consequences of the flexible exploitation of such researcher degrees of freedom are better understood [17,18], one might have expected this to translate into corrections mentioning questionable research practices or '*p*-hacking'. However, that was not the case in any of the corrections we examined. Indeed, retrospective disclosure of this nature is rare in general (see also Rohrer *et al.* [19]).

Moreover, even if errors are discovered or recognized, there is no guarantee that this will result in a correction. Consider, for instance, the aforementioned study by Hardwicke *et al.* [3]. Of the 24 articles containing irreproducible results, only three author teams proposed to publish a correction, and, to date, only one has effectively been published. That said, Hardwicke and colleagues noted that the inability to reproduce those particular results seemingly did not affect the original papers' conclusions. Hence, authors and/or journals might find the publication of a formal correction unnecessary in such cases where the mistakes do not have a big impact on the conclusions. This is also reflected in the policy of *Psychological Science* ('2020 Submission Guidelines', n.d. [20]), which states that:

> A correction notice will be published if an error affects the publication record, the scientific integrity of the article, or the reputation of the authors or the journal. In general, *Psychological Science* will not publish a formal correction for spelling or grammatical errors or for errors that do not significantly affect an article's findings or conclusions or a reader's understanding.

Similarly, the American Psychological Association state in their code of conduct that 'If psychologists discover significant errors in their published data, they take reasonable steps to correct such errors in a correction, retraction, erratum, or other appropriate publication means' ('Ethical Principles of Psychologists and Code of Conduct' [21]).

Leaving it up to the authors to evaluate when an error is deemed 'significant' makes sense because of their obvious familiarity with the work. However, having to admit a mistake, especially one that impacts the conclusions of their paper, might be considered embarrassing or potentially detrimental to one's career. Furthermore, cognitive biases and motivated reasoning [22] may play a role in deciding whether an error is significant or not. Researchers are human too, which not only means that they make mistakes but also that their reactions to mistakes can be imperfect. Therefore, the individual self-correction policies quoted above may be viewed as too naive.

Interestingly, other guidelines like the one from the International Committee of Medical Journal Editors are more straightforward (International Committee of Medical Journal Editors, [23]):

> Pervasive errors can result from a coding problem or a miscalculation and may result in extensive inaccuracies throughout an article. If such errors do not change the direction or significance of the results, interpretations, and conclusions of the article, a correction should be published…Errors serious enough to invalidate a paper's results and conclusions may require retraction. However, retraction with republication (also referred to as 'replacement') can be considered in cases where honest error (e.g. a misclassification or miscalculation) leads to a major change in the direction or significance of the results, interpretations, and conclusions.

It is noteworthy that manuscripts submitted to *Psychological Science* should 'conform to the Recommendations for the Conduct, Reporting, Editing and Publication of Scholarly Work in Medical Journals, which can be found in full at www.icmje.org' ('2020 Submission Guidelines', n.d. [20]). Hence, it is remarkable that *Psychological Science's* policy 'to not publish a formal correction … for errors that do not significantly affect an article's findings or conclusions' contradict the recommendation from the International Committee of Medical Journal Editors to publish a correction when errors 'do not change the direction or significance of the results, interpretations and conclusions of the article.' Their policy might explain why correction notices in *Psychological Science* more often mentioned mistakes in the data-analysis and involved pertinent results, whereas issues regarding metadata were less prevalent compared with the other two journals. Of course, it is difficult to attribute this apparent difference (exclusively) to the journals' policies. Alternatively, the editorial/copyediting process could be more rigorous at *Psychological Science* leading to fewer metadata mistakes, and/or their articles might be more likely to draw attention from other researchers looking to critically examine the data-analysis, for example.

## 4.2. Glass half-empty?

The current situation is less than ideal for a number of reasons. First, one might argue that there is a bias in the type of corrections that get published. Errors that could change the conclusions might end up in the file drawer, for instance, because journals implicitly discourage the publication of such corrections, or as a result of the (perceived) stigma surrounding the admission of mistakes. In other cases, seemingly fundamental alterations (e.g. correlations reversing signs, switching constructs due to errant coding, and the like) get portrayed as minor changes that do not impact the conclusions. Alternatively, it might mean that the findings in question are very robust, or that the corresponding theories are unfalsifiable to begin with.

Second, we happened to notice that some corrections still contained errors or even introduced new ones. This seems to indicate that there is little reviewing involved, and that they are seen as a formality or a nuisance, rather than a source of information for readers.

Third, in some cases, authors themselves discover an error in their work, but it is not uncommon for researchers to stumble upon an error in someone else's paper. Our results show that very few corrections actually mention or thank a third party, by name or anonymously, for uncovering the error(s). This might merely reflect a low prevalence of such instances. However, we know from anecdotes on social media and personal experience that some corrections in our sample did not acknowledge the researcher(s) who discovered the error(s). Consequently, the true rate of such instances might be higher. Note though that in those cases both the authors and the journal do not follow the guidelines regarding publication credit. For instance, the code of conduct of the APA states: 'Minor contributions to the research or to the writing for publications are acknowledged appropriately, such as in footnotes or in an introductory statement.' It is unclear why these guidelines would not apply to correction notices, except perhaps those involving trivial changes (e.g. the spelling of an author's name). Indeed, one could even make the case for coauthorship as some instances pertain '[s]ubstantial contributions to … the analysis or interpretation of data for the work' (International Committee of Medical Journal Editors, [23]), provided that other criteria such as approval of the final version, and accountability for the work are also fulfilled.

A last issue, related to the previous point, concerns the procedure of handling mistakes spotted by other researchers. The obvious first step is to contact the original authors. But what if they do not respond, or acknowledge the mistake vowing to inform the journal, yet nothing seems to happen? As far as we know, there are very few recommendations to streamline this process. *Frontiers* does have a general complaints policy, which also covers issues about the scientific validity of a paper. More specifically, they advise the 'complainant' to get in touch with the authors first, and to only contact

the journal if the authors do not respond or take no further action ('Publishing Model', n.d. [24]). In addition, they require a detailed description of any exchanges with the authors as well as 'an annotated PDF of the article … that clearly marks the passages concerned and the reasons why they are of concern.' Such policies arguably encourage researchers to remain silent about mistakes they discovered in someone else's work.

## 4.3. Or glass half-full?

There is definitely room for improvement in terms of journal policies and researchers' attitudes regarding individual self-correction. However, there is also reason for optimism. For instance, Rohrer *et al*. [19] initiated the loss-of-confidence project in which they call for researchers to disclose their doubts about previously published work. Such initiatives can help alleviate the stigma surrounding individual self-correction, and thus provide the impetus for their (re)appreciation. In this regard, it is noteworthy that replication studies, another form of scientific self-correction, were very unpopular within psychology (Makel *et al*. [25]), but are gradually becoming more mainstream in recent years [13]. Perhaps, we might see a similar trend for (non-trivial) corrections. After all, they offer an important and efficient way to progress science as a whole. We hope that by raising awareness regarding their potential and pointing out some shortcomings, both researchers and journals will aim to optimize the content and guidelines pertaining correction notices.

## 4.4. Prevention is better than cure

Rather than having to correct mistakes, we, as a field, should also strive to *avoid* mistakes. In this final section, we discuss some concrete approaches researchers and journals could take to achieve that goal. This is by no means an exhaustive list, and we also do not discuss potential macro-level initiatives (e.g. changing the publish-or-perish culture might give researchers more breathing room to carry out and check their work (even) more rigorously).

A substantial portion of errors in our sample involved misreporting statistical output. For instance, one correction notice stated that inconsistencies arose 'because of errors copying values from previous drafts, missed lines in the SPSS output, and reference to outdated datasets and analyses during a protracted write-up process' [26, p. 1]. Such mistakes can be averted to a considerable extent by adopting (a combination of) sound practices like keeping a well-documented data-management plan, and/or using software to integrate one's analysis code into their manuscript (e.g. rmarkdown or jupyter notebooks). In addition, one could work with a 'co-pilot'; someone, often a co-author, who independently verifies data-handling, data-analysis, reporting etc. (e.g. [27]). One could also use tools such as statcheck [4] to ascertain that (certain aspects of) the results are properly reported. In fact, some journals, including *Psychological Science*, run all to-be-published articles through statcheck or require authors to provide a clean statcheck report themselves ('2020 Submission Guidelines', n.d. [20]). However, high-quality journals could (or should) go further in their efforts to properly vet prospective manuscripts. They could appoint one or more specialists to thoroughly verify the reproducibility of a study's outcomes as a (final) part of the review process (see also Sakaluk *et al*. [28]). This is for instance the case for articles published in *Biometrical Journal*, where accepted articles' data and code get reviewed, either by the so-called Reproducible Research Editor or an external reviewer. According to a report by Hofner *et al*. [29], this takes an average of 54 days, often including one or more rounds of revision, which is somewhat comparable to the proofreading process, except that permanent reproducibility issues would prompt a reassessment of the work.

Currently, some reviewers take it upon themselves to perform a reproducibility check, which can be very time-consuming, and arguably goes beyond what one ought to expect from a voluntary service to the field. This is where journals could play an important role, thereby also addressing questions regarding their added value, since technological advancement has rendered their dissemination and type-setting role quasi obsolete. Moreover, as the rate of data sharing will (presumably) increase (further), for instance, due to journal policies (e.g. [30]), so will the potential to uncover previously undetectable errors (e.g. Hardwicke *et al.* [3]). It is therefore in a journal's best interest to avoid having to issue correction notices on several of its published articles.

Interestingly, a recent survey among editors of psychology journals indicated that specialized statistical review, defined as the technical assessment provided by statistical experts regarding the analysis and presentation of quantitative information, rarely occurs and is often regarded as unnecessary [31]. Furthermore, in response to Hardwicke *et al*.'s [3] findings regarding data

reusability and analytical reproducibility (see above), the editors of *Cognition* issued a commentary [32] saying that:

> Editors and associate editors check that data are made available and in a readable format, but editors and reviewers cannot take on the responsibility of checking for analytical reproducibility. Ultimately it is the authors who are responsible for submitting accurate data to the journal.

In time, journals might change their stance, and follow the example of other fields like biomedicine, where the (advantage of) in-depth statistical review is more generally recognized [31]. Within psychology, we only know of one journal, *Meta-Psychology*, with such a policy. Interestingly, Stephen Lindsay, former editor-in-chief of *Psychological Science*, suggested, in his so-called swan song editorial, that journals and scientific societies should add value to their publications by investing more resources [33]. Offering independent verification of analytic reproducibility would be an excellent step in this regard.

Data accessibility. The manuscript was written in R [34] using the packages papaja [35] and rmarkdown [36]. On the project's page (https://osf.io/8ec57/), one can find the .Rmd file, the materials and the data. The analysis code is available in a Code Ocean container (https://doi.org/10.24433/CO.3352846.v1). We report how we determined our sample size, all data exclusions, all manipulations and all measures in the study.

Authors' contributions. T.H. developed the design of the study. Both authors coded the corrections, analysed the data, wrote the manuscript and approved the final version for submission.

Competing interests. The authors declare that there were no conflicts of interest with respect to the authorship or the publication of this article.

Funding. Part of this work was conducted while T.H. was a postdoctoral fellow of the Research Foundation-Flanders (FWO-Vlaanderen). A.-S.M. was funded by the KU Leuven Research Counsil grant no. C14/16032.

Acknowledgements. We thank Mark de Rooij, Pieter Moors, Julia Rohrer, Gert Storms and Wolf Vanpaemel for providing feedback on an earlier version of the manuscript.

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
