## [Reviewer comments · Royal Society Open Science]

Review History

RSOS-200834.R0 (Original submission)

Review form: Reviewer 1

Is the manuscript scientifically sound in its present form?

Yes

Are the interpretations and conclusions justified by the results?

Yes

Is the language acceptable?

Yes

Do you have any ethical concerns with this paper?

No

Have you any concerns about statistical analyses in this paper?

No

Recommendation?

Major revision is needed (please make suggestions in comments)

Comments to the Author(s)

Manuscript ID: RSOS-200834

Manuscript title: Corrections in psychology: Impactful or inconsequential?

Summary

This article reports a study with a cross-sectional design intended to examine the content of correction notices in three psychology journals published between 2010 and 2018. The article is generally clearly written and concise (though the discussion is a little lengthy and at times speculates well beyond the data). The conclusions are generally well calibrated to the observed evidence, though I have comments on some specific claims (see below). The studies limitations are mostly acknowledged, though this could be made more explicit (more detail below). The study conforms to excellent standards of transparency in many respects (open data, open analysis script, code interleaved with prose to render the final paper), though it is not pre-registered. Overall, I believe the study makes an informative contribution to the literature. Below I have outlined some major comments that I feel should be addressed prior to acceptance and some minor comments/suggestions.

Major Comments

- The abstract would be more informative if it clearly stated the research design, how the sample was obtained, and the sample size.
- Due to feasibility constraints, the majority of the analyses pertain to only three psychology journals that had published the most correction notices in the time period of interest. This is fine – however I think it should be acknowledged that this is a limitation of the study that will introduce bias (to the extent that one wishes to generalize beyond the three journals, which the title of the manuscript implies is an aim).
- As far as I can see, the study was not pre-registered. It is not necessarily problematic for a study to not be pre-registered, but pre-registration can help to reduce bias. Therefore, this study has a higher risk of bias than if it had been pre-registered and it might be informative to explicitly acknowledge that. It is helpful that the authors have included this statement “We report how we determined our sample size, all data exclusions, all manipulations, and all measures in the study.” – though this appears in section called “Author-supplied statements” so I’m not sure if that means it will be in the manuscript itself (I think it should be).
- I’m a bit unclear on the determination of whether changes in results impacted the original conclusions. The manuscript states that the implications of results were categorized according to whether they ‘bolstered’ the original conclusion which is defined as “values more extreme in the direction of the drawn conclusion versus values being less extreme or going in the opposite direction of what was claimed initially”. Does that mean that if a hypothesis predicted a positive effect, the original, say, Cohen’s d , was 0.6 and the correction said it was actually 0.5, this would be classified as ‘not bolstering the original hypothesis’ because the value is ‘less extreme’? And if the corrected Cohen’s d was -0.2, this would also be classified as ‘not bolstering the original hypothesis’ because the effect is in the opposite direction? If that is how the classification was performed then it seems problematic to me to conflate these two scenarios. In the first, the Cohen’s d is smaller than originally supported, but would still be consistent with the original directional hypothesis. In the second, the corrected value not only doesn’t ‘bolster’ the original hypothesis, it contradicts it.
- Related to the above, footnote 2 suggests that “hypotheses tested within psychology are often somewhat vague in that they merely specify an effect in a given direction.” I fear it may be worse than that – isn’t it the case that many hypotheses tested in psychology are in fact non-directional (i.e., they state there will be an effect but are agnostic as to the direction)? I’m not sure it is justified to assume that most are directional without empirical data to back that up.
- P10 L10-12 “we can conclude that a substantial fraction does not impact or update the scientific knowledge base” – I think this claim should perhaps be rephrased because a correction notice by definition updates the scientific knowledge base. Do the authors mean that the

correction notices they examined do not have a meaningful impact of scientific conclusions? If so, that could be made clearer.

- P10 L15-18 “funding information (which, in most psychology studies, has no bearing on the outcome)”. I think that strong statement needs empirical support. Funding sources can indicate a risk of bias, and I can’t see why they wouldn’t be pertinent in psychology studies just as they are in medical studies.

- P10 L19-21 “one might argue that science as a whole benefits very little from these forms of individual self-correction.” That may be the authors opinion, but it seems hard to justify on the basis of the data collected, so I think more argumentation is needed. How do we know the corrections had no benefit? If they had no benefit, why would anybody bother making the correction? Or even getting things right in the first place? Apparently minor corrections could have important impacts. For example, changes to quantitative values that do not appear to affect conclusions could still have an effect on e.g., meta-analyses or attempts to replicate. Changes to author names or ordering could impact on careers. Changes to funding statements could affect risk of bias assessments.

- P13 L15 – 19 “For instance, Hardwicke et al. (2018) found that 13 out of 35 examined articles published with open data in the journal *Cognition* contained at least one value that could not be reproduced.” The finding reported is correct, however it is the reproducibility outcome *after author assistance*. More pertinent here I think would be the reproducibility outcome *before author assistance*, as all of those reproducibility issues would arguably require publication of a correction. Before author assistance, there were reproducibility issues in 24 of the 35 examined articles.

- It would be interesting to see a breakdown of the results by journal. In the discussion, it is pointed out that *Psychological Science* has an explicit policy of not publishing corrections unless they are consequential to the study’s conclusions. Is that also apparent in the data?

Minor Comments

- One suggestion to ensure the long-term reproducibility of the manuscript is to use a service like Code Ocean or a tool like Binder to preserve the software environment in which the analyses run successfully. This helps to avoid problems when others cannot run your code due to dependency issues.

- On what date was the Scopus search performed?

- The data on number of corrections published in psychology shown in figure 1 is presented in a rather auxiliary fashion as part of the methods section – however, to me it seems sufficiently interesting to promote to the results section – particularly the finding that “there is about 1 correction per 100 articles published within psychology.”

- Footnote 3 mentions that there were 16 retraction notices in the sample – could it be clarified whether (a) the sampling procedure was sensitive to all retraction notices (i.e., were there really only 16 total retractions at the three journals over the time period of interest?) and (b) were retraction notices included in the analyses reported in this paper?

- The title could be made clearer. Specifically, (1) to be more informative it should ideally mention the design of study; (2) to be clearer, it should ideally refer to ‘correction notices’ rather than just ‘correction’ – as the authors point out, there are many different types of correction, so the current title is ambiguous; (3) The part of the title “in psychology” is rather overstating the sample as only three journals were examined. To be clear – I think three journals is informative and useful, I just think the current title is a bit ambiguous and potentially misleading.

- P5 L36 – “the perceived impact on the paper’s initial conclusions” – this could be clearer, who’s perception is being referred to?

- In table 1, missing word before ‘else’ – someone else?

- P7 L15 “a rest category” – does this mean an “other category” ?

- The text says “Figure 2 illustrates the percentage of corrected findings” however if I’ve understood correctly I think that’s an inaccurate description – it shows the distribution of correction notice types, some of which involve corrections to the findings and some of which don’t (e.g., changes to metadata).

- Perhaps some examples could be provided to illustrate the different correction types e.g., what would be an example of a correction related to ‘aesthetics’?

- It would be interesting to know how many correction notices presented more than one correction and more than one correction type. For example, a correction notice that relates to single typo is quite different to one that has to correct ten typos, and different again to one that corrects ten typos, a hypothesis, and several aspects of the methodology.
- Given the relatively small sample size, it would be helpful if percentages were accompanied by the relevant counts (numerators). The relevant denominators could be made a bit clearer too (e.g., "Of the X corrections that involved changes to the results, X (X%) were...")
- P9 L12 - "we adapted" should this say "we adopted"? "adapted" implies the method was modified but as far as I can tell the same method was used.
- The percentage error metric describe around P9 L12 should perhaps be reported in the methods section rather than the results?
- P15 L12-15 "It is noteworthy that manuscripts submitted to Psychological Science should also conform to these guidelines." - could it be clarified why? Is PS a member of ICMJE?
- P15 L48-49 "Second, some of the corrections we reviewed still contained errors or even introduced new ones." This is mentioned in the discussion but I don't recall seeing any results related to the introduction of new errors in the corrections themselves - apologies if I missed it.
- P16 L7-9 "However, we know that some corrections in our sample did not acknowledge the researcher(s) who discovered the error(s)." What is the basis for this statement?
- P16 L17-21 "In this regard, it is noteworthy that replication studies, another form of scientific self-correction, were very unpopular within psychology (Makel, Plucker, & Hegarty, 2012), but are gradually becoming more mainstream in recent years." This statement needs empirical support. A more recent estimate is available here:
<https://doi.org/10.31222/osf.io/9sz2y>
- P17 L48-50 "A substantial portion of errors in our sample involved misreporting statistical output (e.g., due to a protracted review process)." It's not clear to me why a protracted review process would lead to misreporting statistical output.
- P17 L10-12 "In fact, some journals, including Psychological Science, run all to-be-published articles through statcheck by default." I believe they now just ask authors to do this themselves and self-certify the manuscript passed.
- P17 L15-18 "They could appoint one or more specialists to thoroughly verify the reproducibility of a study's outcomes as a (final) part of the review process." Sakulak et al. (2014) discuss this at length so it might be a citation of interest:
<https://doi.org/10.1177%2F1745691614549257>

Review form: Reviewer 2

Is the manuscript scientifically sound in its present form?

No

Are the interpretations and conclusions justified by the results?

No

Is the language acceptable?

Yes

Do you have any ethical concerns with this paper?

No

Have you any concerns about statistical analyses in this paper?

Yes

Recommendation?

Reject

Comments to the Author(s)

This paper provided a content analysis of corrections published in three journals from 2010 to 2018 (Frontiers in Psychology, $n = 99$; Journal of Affective Disorders, $n = 57$; Psychological Science, $n = 58$). The main conclusion was that “a substantial fraction [of corrections] do not impact or update the scientific knowledge base.” The implication is that the existence of corrections is not likely to be an efficient or an effective means by which science is self-correcting. The authors suggest that formal steps such as analytic review (e.g., Sakaluk et al., 2014) are needed to avoid mistakes are likely to be a more successful for producing more robust journal articles.

Bottom Line: I don't disagree with many of the suggestions in the paper. However, I am not sure the content analysis here is necessary to support those suggestions.

I think the current study needs stronger motivation and many of the analytic decisions need more exposition. Some of the choices may undercut the contribution to the point that it difficult to draw conclusions from the work. The big issues to me where the seemingly ad hoc selection of years and the restriction to just three outlets. In some ways, I think it would be stronger to randomly select corrections from each year and also code journal characteristics by things like policies, field, etc. As it stands, I am not sure what to conclude from what is reported.

Specifics

1. The paper would benefit from a stronger motivation. What is present in the short introduction hinges on whether there is validity for the proposition that the existence of corrections indicates that science is self-correcting. More from the Grear (2013) piece could help make this case. To be honest, I don't think I would argue that the existence of corrections in journals is especially strong evidence one way or the other.
2. The justification for using Scopus needs to be stronger. More details about the initial dataset should be provided including the total number of articles published in each journal and what those time trends looked like from 2010 to 2018. I think that statistic is more informative than just the number of corrections divided by the number of articles published in Psychology as was depicted in Figure 1. Journal specific trends would be more informative.
3. I think stronger justification for examining the years of 2010 to 2018 and for picking just those three journals needs to be in the paper. What was provided might not strike all readers as convincing.
4. One concern is that different journals/societies may have different norms/expectations about corrections. The authors correctly discuss this. However, I think this has analytic consequences in that corrections are nested within journals with time as a further factor. So, I think a complicated multi-level structure is at play in terms of what is likely to be the data generation mechanism. This issue might be worth considering.
5. Related to the points about the three outlets, more details about the journals including their policies and how many times editorial teams turned over could be illuminating.
6. The initial agreement on the coding scheme should be reported and discussed. It seemed like the percentage agreement for whether a result is judged to bolster the original conclusion was low (52.63% and I don't think this was corrected for chance agreement). This kind of discussion is important for interpreting the results (i.e., it speaks to the reliability of the coding). The 53% initial figure raised some concerns for me.
7. Some justification for the PE ratio over 10 being classified as a major error should be provided in the report.

Decision letter (RSOS-200834.R0)

Dear Dr Heyman,

The editors assigned to your paper ("Corrections in psychology: Impactful or inconsequential?") have now received comments from reviewers. We would like you to revise your paper in accordance with the referee and Associate Editor suggestions which can be found below (not including confidential reports to the Editor). Please note this decision does not guarantee eventual acceptance.

Please submit a copy of your revised paper before 19-Aug-2020. Please note that the revision deadline will expire at 00.00am on this date. If we do not hear from you within this time then it will be assumed that the paper has been withdrawn. In exceptional circumstances, extensions may be possible if agreed with the Editorial Office in advance. We do not allow multiple rounds of revision so we urge you to make every effort to fully address all of the comments at this stage. If deemed necessary by the Editors, your manuscript will be sent back to one or more of the original reviewers for assessment. If the original reviewers are not available, we may invite new reviewers.

- Data accessibility

If you wish to submit your supporting data or code to Dryad (<http://datadryad.org/>), or modify your current submission to dryad, please use the following link:
<http://datadryad.org/submit?journalID=RSOS&manu=RSOS-200834>

- Competing interests

- Authors' contributions

- Acknowledgements

- Funding statement

on behalf of Professor Zoltan Dienes (Associate Editor) and Essi Viding (Subject Editor)
openscience@royalsociety.org

Associate Editor's comments (Professor Zoltan Dienes):

The reviewers have made many useful and detailed suggestions; please show how you have addressed each of them in your revision.

Reviewers' Comments to Author:

Reviewer: 1

Comments to the Author(s)

Manuscript ID: RSOS-200834

Manuscript title: Corrections in psychology: Impactful or inconsequential?

Summary

This article reports a study with a cross-sectional design intended to examine the content of correction notices in three psychology journals published between 2010 and 2018. The article is generally clearly written and concise (though the discussion is a little lengthy and at times speculates well beyond the data). The conclusions are generally well calibrated to the observed evidence, though I have comments on some specific claims (see below). The studies limitations are mostly acknowledged, though this could be made more explicit (more detail below). The study conforms to excellent standards of transparency in many respects (open data, open analysis script, code interleaved with prose to render the final paper), though it is not pre-registered. Overall, I believe the study makes an informative contribution to the literature. Below I have outlined some major comments that I feel should be addressed prior to acceptance and some minor comments/suggestions.

Major Comments

- The abstract would be more informative if it clearly stated the research design, how the sample was obtained, and the sample size.
- Due to feasibility constraints, the majority of the analyses pertain to only three psychology journals that had published the most correction notices in the time period of interest. This is fine – however I think it should be acknowledged that this is a limitation of the study that will introduce bias (to the extent that one wishes to generalize beyond the three journals, which the title of the manuscript implies is an aim).
- As far as I can see, the study was not pre-registered. It is not necessarily problematic for a study to not be pre-registered, but pre-registration can help to reduce bias. Therefore, this study has a higher risk of bias than if it had been pre-registered and it might be informative to explicitly acknowledge that. It is helpful that the authors have included this statement “We report how we determined our sample size, all data exclusions, all manipulations, and all measures in the study.” – though this appears in section called “Author-supplied statements” so I’m not sure if that means it will be in the manuscript itself (I think it should be).
- I’m a bit unclear on the determination of whether changes in results impacted the original conclusions. The manuscript states that the implications of results were categorized according to whether they ‘bolstered’ the original conclusion which is defined as “values more extreme in the direction of the drawn conclusion versus values being less extreme or going in the opposite direction of what was claimed initially”. Does that mean that if a hypothesis predicted a positive effect, the original, say, Cohen’s d , was 0.6 and the correction said it was actually 0.5, this would be classified as ‘not bolstering the original hypothesis’ because the value is ‘less extreme’? And if the corrected Cohen’s d was -0.2, this would also be classified as ‘not bolstering the original hypothesis’ because the effect is in the opposite direction? If that is how the classification was performed then it seems problematic to me to conflate these two scenarios. In the first, the Cohen’s d is smaller than originally supported, but would still be consistent with the original directional hypothesis. In the second, the corrected value not only doesn’t ‘bolster’ the original hypothesis, it contradicts it.
- Related to the above, footnote 2 suggests that “hypotheses tested within psychology are often somewhat vague in that they merely specify an effect in a given direction.” I fear it may be worse than that – isn’t it the case that many hypotheses tested in psychology are in fact non-directional (i.e., they state there will be an effect but are agnostic as to the direction)? I’m not sure it is justified to assume that most are directional without empirical data to back that up.
- P10 L10-12 “we can conclude that a substantial fraction does not impact or update the scientific knowledge base” – I think this claim should perhaps be rephrased because a correction notice by definition updates the scientific knowledge base. Do the authors mean that the correction notices they examined do not have a meaningful impact of scientific conclusions? If so, that could be made clearer.
- P10 L15-18 “funding information (which, in most psychology studies, has no bearing on

the outcome)". I think that strong statement needs empirical support. Funding sources can indicate a risk of bias, and I can't see why they wouldn't be pertinent in psychology studies just as they are in medical studies.

- P10 L19-21 "one might argue that science as a whole benefits very little from these forms of individual self-correction." That may be the authors opinion, but it seems hard to justify on the basis of the data collected, so I think more argumentation is needed. How do we know the corrections had no benefit? If they had no benefit, why would anybody bother making the correction? Or even getting things right in the first place? Apparently minor corrections could have important impacts. For example, changes to quantitative values that do not appear to affect conclusions could still have an effect on e.g., meta-analyses or attempts to replicate. Changes to author names or ordering could impact on careers. Changes to funding statements could affect risk of bias assessments.

- P13 L15 - 19 "For instance, Hardwicke et al. (2018) found that 13 out of 35 examined articles published with open data in the journal *Cognition* contained at least one value that could not be reproduced." The finding reported is correct, however it is the reproducibility outcome *after author assistance*. More pertinent here I think would be the reproducibility outcome *before author assistance*, as all of those reproducibility issues would arguably require publication of a correction. Before author assistance, there were reproducibility issues in 24 of the 35 examined articles.

- It would be interesting to see a breakdown of the results by journal. In the discussion, it is pointed out that Psychological Science has an explicit policy of not publishing corrections unless they are consequential to the study's conclusions. Is that also apparent in the data?

Minor Comments

- One suggestion to ensure the long-term reproducibility of the manuscript is to use a service like Code Ocean or a tool like Binder to preserve the software environment in which the analyses run successfully. This helps to avoid problems when others cannot run your code due to dependency issues.

- On what date was the Scopus search performed?

- The data on number of corrections published in psychology shown in figure 1 is presented in a rather auxiliary fashion as part of the methods section - however, to me it seems sufficiently interesting to promote to the results section - particularly the finding that "there is about 1 correction per 100 articles published within psychology."

- Footnote 3 mentions that there were 16 retraction notices in the sample - could it be clarified whether (a) the sampling procedure was sensitive to all retraction notices (i.e., were there really only 16 total retractions at the three journals over the time period of interest?) and (b) were retraction notices included in the analyses reported in this paper?

- The title could be made clearer. Specifically, (1) to be more informative it should ideally mention the design of study; (2) to be clearer, it should ideally refer to 'correction notices' rather than just 'correction' - as the authors point out, there are many different types of correction, so the current title is ambiguous; (3) The part of the title "in psychology" is rather overstating the sample as only three journals were examined. To be clear - I think three journals is informative and useful, I just think the current title is a bit ambiguous and potentially misleading.

- P5 L36 - "the perceived impact on the paper's initial conclusions" - this could be clearer, who's perception is being referred to?

- In table 1, missing word before 'else' - someone else?

- P7 L15 "a rest category" - does this mean an "other category" ?

- The text says "Figure 2 illustrates the percentage of corrected findings" however if I've understood correctly I think that's an inaccurate description - it shows the distribution of correction notice types, some of which involve corrections to the findings and some of which don't (e.g., changes to metadata).

- Perhaps some examples could be provided to illustrate the different correction types e.g., what would be an example of a correction related to 'aesthetics'?

- It would be interesting to know how many correction notices presented more than one correction and more than one correction type. For example, a correction notice that relates to

single typo is quite different to one that has to correct ten typos, and different again to one that corrects ten typos, a hypothesis, and several aspects of the methodology.

- Given the relatively small sample size, it would be helpful if percentages were accompanied by the relevant counts (numerators). The relevant denominators could be made a bit clearer too (e.g., "Of the X corrections that involved changes to the results, X (X%) were...")

- P9 L12 – "we adapted" should this say "we adopted"? "adapted" implies the method was modified but as far as I can tell the same method was used.

- The percentage error metric describe around P9 L12 should perhaps be reported in the methods section rather than the results?

- P15 L12-15 "It is noteworthy that manuscripts submitted to Psychological Science should also conform to these guidelines." – could it be clarified why? Is PS a member of ICMJE?

- P15 L48-49 "Second, some of the corrections we reviewed still contained errors or even introduced new ones." This is mentioned in the discussion but I don't recall seeing any results related to the introduction of new errors in the corrections themselves – apologies if I missed it.

- P16 L7-9 "However, we know that some corrections in our sample did not acknowledge the researcher(s) who discovered the error(s)." What is the basis for this statement?

- P16 L17-21 "In this regard, it is noteworthy that replication studies, another form of scientific self-correction, were very unpopular within psychology (Makel, Plucker, & Hegarty, 2012), but are gradually becoming more mainstream in recent years." This statement needs empirical support. A more recent estimate is available here: <https://doi.org/10.31222/osf.io/9sz2y>

- P17 L48-50 "A substantial portion of errors in our sample involved misreporting statistical output (e.g., due to a protracted review process)." It's not clear to me why a protracted review process would lead to misreporting statistical output.

- P17 L10-12 "In fact, some journals, including Psychological Science, run all to-be-published articles through statcheck by default." I believe they now just ask authors to do this themselves and self-certify the manuscript passed.

- P17 L15-18 "They could appoint one or more specialists to thoroughly verify the reproducibility of a study's outcomes as a (final) part of the review process." Sakulak et al. (2014) discuss this at length so it might be a citation of interest: <https://doi.org/10.1177%2F1745691614549257>

Reviewer: 2

Comments to the Author(s)

This paper provided a content analysis of corrections published in three journals from 2010 to 2018 (Frontiers in Psychology, n = 99; Journal of Affective Disorders, n = 57; Psychological Science, n = 58). The main conclusion was that "a substantial fraction [of corrections] do not impact or update the scientific knowledge base." The implication is that the existence of corrections is not likely to be an efficient or an effective means by which science is self-correcting. The authors suggest that formal steps such as analytic review (e.g., Sakaluk et al., 2014) are needed to avoid mistakes are likely to be a more successful for producing more robust journal articles.

Bottom Line: I don't disagree with many of the suggestions in the paper. However, I am not sure the content analysis here is necessary to support those suggestions.

I think the current study needs stronger motivation and many of the analytic decisions need more exposition. Some of the choices may undercut the contribution to the point that it difficult to draw conclusions from the work. The big issues to me where the seemingly ad hoc selection of years and the restriction to just three outlets. In some ways, I think it would be stronger to randomly select corrections from each year and also code journal characteristics by things like policies, field, etc. As it stands, I am not sure what to conclude from what is reported.

Specifics

1. The paper would benefit from a stronger motivation. What is present in the short introduction hinges on whether there is validity for the proposition that the existence of corrections indicates

that science is self-correcting. More from the Grear (2013) piece could help make this case. To be honest, I don't think I would argue that the existence of corrections in journals is especially strong evidence one way or the other.

2. The justification for using Scopus needs to be stronger. More details about the initial dataset should be provided including the total number of articles published in each journal and what those time trends looked like from 2010 to 2018. I think that statistic is more informative than just the number of corrections divided by the number of articles published in Psychology as was depicted in Figure 1. Journal specific trends would be more informative.

3. I think stronger justification for examining the years of 2010 to 2018 and for picking just those three journals needs to be in the paper. What was provided might not strike all readers as convincing.

4. One concern is that different journals/societies may have different norms/expectations about corrections. The authors correctly discuss this. However, I think this has analytic consequences in that corrections are nested within journals with time as a further factor. So, I think a complicated multi-level structure is at play in terms of what is likely to be the data generation mechanism. This issue might be worth considering.

5. Related to the points about the three outlets, more details about the journals including their policies and how many times editorial teams turned over could be illuminating.

6. The initial agreement on the coding scheme should be reported and discussed. It seemed like the percentage agreement for whether a result is judged to bolster the original conclusion was low (52.63% and I don't think this was corrected for chance agreement). This kind of discussion is important for interpreting the results (i.e., it speaks to the reliability of the coding). The 53% initial figure raised some concerns for me.

7. Some justification for the PE ratio over 10 being classified as a major error should be provided in the report.

Author's Response to Decision Letter for (RSOS-200834.R0)

See Appendix A.

Decision letter (RSOS-200834.R1)

Dear Dr Heyman,

It is a pleasure to accept your manuscript entitled "Correction notices in psychology: Impactful or inconsequential?" in its current form for publication in Royal Society Open Science.

Please ensure that you send to the editorial office an editable version of your accepted manuscript, and individual files for each figure and table included in your manuscript. You can send these in a zip folder if more convenient. Failure to provide these files may delay the

processing of your proof. You may disregard this request if you have already provided these files to the editorial office.

on behalf of Professor Zoltan Dienes (Associate Editor) and Essi Viding (Subject Editor)
openscience@royalsociety.org

Appendix A

To the editors of *Royal Society Open Science*,

Please find below our responses to the comments of the reviewers. We will describe the changes made in response to each suggestion. The comments are numbered per reviewer. Every section will be preceded by the suggestion/question in *italic*. We would like to thank the reviewers for their constructive comments. We hope you agree that the revisions based on their input have led to significant improvements in the manuscript.

Sincerely,

Tom Heyman and Anne-Sofie Maerten

Reviewer 1

Major points

1. The abstract would be more informative if it clearly stated the research design, how the sample was obtained, and the sample size.

The abstract now describes the sample size, how it was obtained (i.e., via Scopus), how the three particular journals were selected, and that it concerns exploratory analyses.

2. Due to feasibility constraints, the majority of the analyses pertain to only three psychology journals that had published the most correction notices in the time period of interest. This is fine – however I think it should be acknowledged that this is a limitation of the study that will introduce bias (to the extent that one wishes to generalize beyond the three journals, which the title of the manuscript implies is an aim).

On p. 6 we mention that this is a non-random sample, and on p. 14 we explicitly state that “it remains to be seen whether these results generalize to other journals within psychology.”

3. As far as I can see, the study was not pre-registered. It is not necessarily problematic for a study to not be pre-registered, but pre-registration can help to reduce bias. Therefore, this study has a higher risk of bias than if it had been pre-registered and it might be informative to explicitly acknowledge that. It is helpful that the authors have included this statement “We report how we determined our sample size, all data exclusions, all manipulations, and all

measures in the study.” – though this appears in section called “Author-supplied statements” so I’m not sure if that means it will be in the manuscript itself (I think it should be).

The statement was included under the section titled “Data, code and materials” after the discussion section, and as such is (also) part of the manuscript itself. Regarding pre-registration, we added the following on p. 8: “Note that the current study did not seek to test any a priori hypotheses. Instead, it was exploratory in nature, as little is known about this domain. Also, the study’s protocol was not formally pre-registered, even though the coding scheme was designed in advance. As such, the results should be interpreted with due caution.”

4. I’m a bit unclear on the determination of whether changes in results impacted the original conclusions. The manuscript states that the implications of results were categorized according to whether they ‘bolstered’ the original conclusion which is defined as “values more extreme in the direction of the drawn conclusion versus values being less extreme or going in the opposite direction of what was claimed initially”. Does that mean that if a hypothesis predicted a positive effect, the original, say, Cohen’s d , was 0.6 and the correction said it was actually 0.5, this would be classified as ‘not bolstering the original hypothesis’ because the value is ‘less extreme’? And if the corrected Cohen’s d was -0.2, this would also be classified as ‘not bolstering the original hypothesis’ because the effect is in the opposite direction? If that is how the classification was performed then it seems problematic to me to conflate these two scenarios. In the first, the Cohen’s d is smaller than originally supported, but would still be consistent with the original directional hypothesis. In the second, the corrected value not only doesn’t ‘bolster’ the original hypothesis, it contradicts it.

This is indeed how we did it, but in addition to giving a qualitative label (bolster, not bolster or unclear), we also looked at it quantitatively (i.e., the percentage error metric for effect sizes). One could indeed envision different cutoffs with different qualitative labels, each with their own advantages and disadvantages. If one were to opt for a further distinction between not-bolster and contradict, where does one place the boundary? Using the example above, how would one classify a Cohen’s d of 0.0, or 0.01, for instance? Or what if the corrected 95% CI was [-1.2; 0.8] does that contradict the original hypothesis? Our classification avoids these issues, but it is admittedly cruder and lumps together different types of corrections. However, this is offset to certain extent by the quantitative evaluation (see Figure 5 and 6).

5. Related to the above, footnote 2 suggests that “hypotheses tested within psychology are often somewhat vague in that they merely specify an effect in a given direction.” I fear it may be worse than that – isn’t it the case that many hypotheses tested in psychology are in fact non-directional (i.e., they state there will be an effect but are agnostic as to the direction)? I’m not sure it is justified to assume that most are directional without empirical data to back that up.

Cho and Abe (2013) found 90.9% directional vs 9.1% non-directional research hypotheses in journals related to marketing research. Lawler and Zimmermann (2019) suggest that such a division is also likely for related disciplines like psychology, but do not offer empirical support for that notion. Note though that two-tailed testing is more prevalent than one-tailed testing, and this misalignment is the topic of both referenced papers. In any case, we revised the footnote as follows: “One could view this as a rather crude measure to evaluate whether a conclusion was bolstered, but theories or hypotheses tested within psychology are often somewhat vague in that they rarely specify the size of an effect a priori.”

6. P10 L10-12 “we can conclude that a substantial fraction does not impact or update the scientific knowledge base” – I think this claim should perhaps be rephrased because a correction notice by definition updates the scientific knowledge base. Do the authors mean that the correction notices they examined do not have a meaningful impact of scientific conclusions? If so, that could be made clearer.

We rephrased the sentence as follows: “we can conclude that a substantial fraction only updates the scientific *record*, but not the underlying knowledge base as such.”

7. P10 L15-18 “funding information (which, in most psychology studies, has no bearing on the outcome)”. I think that strong statement needs empirical support. Funding sources can indicate a risk of bias, and I can’t see why they wouldn’t be pertinent in psychology studies just as they are in medical studies.

We were not able to find any studies on that topic within psychology ourselves (and also asked around). Compared to the huge literature about this topic in the medical sciences, it might indicate psychology does not suffer from this bias (except maybe in related fields involving clinical trials, or more applied research). We still think this statement is accurate for the type of

articles we considered in this study, but we removed it anyway just to err on the side of caution. From memory, none of the corrections involving funding information indicated having a conflict of interest (if it were the case, it should be explicitly mentioned in principle). This could suggest that our statement was accurate.

8. P10 L19-21 “one might argue that science as a whole benefits very little from these forms of individual self-correction.” That may be the authors opinion, but it seems hard to justify on the basis of the data collected, so I think more argumentation is needed. How do we know the corrections had no benefit? If they had no benefit, why would anybody bother making the correction? Or even getting things right in the first place? Apparently minor corrections could have important impacts. For example, changes to quantitative values that do not appear to affect conclusions could still have an effect on e.g., meta-analyses or attempts to replicate. Changes to author names or ordering could impact on careers. Changes to funding statements could affect risk of bias assessments.

Apologies for the confusion. This sentence was not about corrections in general, and it also does not mean they have no benefit. We rephrased it as follows: “That is, a large number of notices concerned author order, the spelling of an author’s name, their affiliation, omitted references, and the like. Even though these specific forms of individual self-correction all rectify a mistake of some kind, they do not revise flawed scientific notions.”

*9. P13 L15 – 19 “For instance, Hardwicke et al. (2018) found that 13 out of 35 examined articles published with open data in the journal Cognition contained at least one value that could not be reproduced.” The finding reported is correct, however it is the reproducibility outcome *after author assistance*. More pertinent here I think would be the reproducibility outcome *before author assistance*, as all of those reproducibility issues would arguably require publication of a correction. Before author assistance, there were reproducibility issues in 24 of the 35 examined articles.*

True, we changed that sentence and one further down that section accordingly.

10. It would be interesting to see a breakdown of the results by journal. In the discussion, it is pointed out that Psychological Science has an explicit policy of not publishing corrections unless they are consequential to the study’s conclusions. Is that also apparent in the data?

We ran the suggested analyses, and indeed there appears to be a difference in the type of corrections published by *Psychological Science* compared to the other two journals. More specifically, *Psychological Science* published relatively more corrections notices mentioning mistakes in the data-analysis, whereas issues regarding metadata were less prevalent compared to the other two journals. Of course, it is difficult to single out the cause of this apparent difference. Perhaps the editorial/copy-editing process is more rigorous at *Psychological Science* leading to fewer metadata mistakes, or their articles are more likely to draw attention from other researchers looking to critically examine the data-analysis, for example. This is now discussed on p. 16-17.

Minor points

11. One suggestion to ensure the long-term reproducibility of the manuscript is to use a service like Code Ocean or a tool like Binder to preserve the software environment in which the analyses run successfully. This helps to avoid problems when others cannot run your code due to dependency issues.

We created a capsule for the project on Code Ocean (see screenshot below). The capsule is kept private and will be made public if the manuscript gets accepted for publication.

12. *On what date was the Scopus search performed?*

The dates are indicated in the captions of Figure 1 and 2 (the coding itself was not conducted on a fixed date).

13. *The data on number of corrections published in psychology shown in figure 1 is presented in a rather auxiliary fashion as part of the methods section – however, to me it seems sufficiently interesting to promote to the results section – particularly the finding that “there is about 1 correction per 100 articles published within psychology.”*

This is indeed a relevant result in its own right, which is also mentioned in the discussion section on p. 14. However, in our paper it is an essential element in determining the sample/inclusion criteria, hence we opted to describe it in the methods section.

14. *Footnote 3 mentions that there were 16 retraction notices in the sample – could it be clarified whether (a) the sampling procedure was sensitive to all retraction notices (i.e., were there really only 16 total retractions at the three journals over the time period of interest?) and (b) were retraction notices included in the analyses reported in this paper?*

No, retractions were not included in the analyses (except for what is mentioned in Footnote 3). This is now explicitly mentioned on p. 8. Cross-referencing with Web of Science revealed no additional results, cross-referencing with the Retraction Watch database revealed one additional retraction. The results returned by Scopus are manually vetted, so the prevalence estimates are lower bounds (the same holds for the corrections).

15. *The title could be made clearer. Specifically, (1) to be more informative it should ideally mention the design of study; (2) to be clearer, it should ideally refer to ‘correction notices’ rather than just ‘correction’ – as the authors point out, there are many different types of correction, so the current title is ambiguous; (3) The part of the title “in psychology” is rather overstating the sample as only three journals were examined. To be clear – I think three journals is informative and useful, I just think the current title is a bit ambiguous and potentially misleading.*

We changed the title to “Correction notices in psychology: Impactful or inconsequential?” It does not address all of the reviewers’ suggestions, but we believe including additional elements

like all the names of the journals would make the title too long. This information is featured prominently in the abstract. A title should draw readers in, without conveying inaccurate information of course. We think the revised title accomplishes that goal. In that sense, it is also comparable to titles of other articles within meta-psychology (e.g., *Estimating the reproducibility of psychological science*, or *The poor availability of psychological research data for reanalysis*).

16. P5 L36 – *“the perceived impact on the paper’s initial conclusions” – this could be clearer, who’s perception is being referred to?*

Both from the authors’ point of view as that of a reader. This was added to the sentence.

17. In table 1, missing word before ‘else’ – someone else?

Yes, we corrected that.

18. P7 L15 *“a rest category” – does this mean an “other category” ?*

Indeed, we clarified this as follows: “rest category including raw effect sizes (e.g., means, unstandardized regression coefficients, and the like; these were labelled as “Other” in Table 1).”

19. The text says “Figure 2 illustrates the percentage of corrected findings” however if I’ve understood correctly I think that’s an inaccurate description – it shows the distribution of correction notice types, some of which involve corrections to the findings and some of which don’t (e.g., changes to metadata).

Yes, that was a mistake, apologies. Because of other changes, this sentence no longer appears in the manuscript, though.

20. Perhaps some examples could be provided to illustrate the different correction types e.g., what would be an example of a correction related to ‘aesthetics’?

We included further details regarding the coding scheme, including examples in a supplementary file on OSF (link:

https://osf.io/uyqra/?view_only=21c1d649126a432d84c6a09ddf16f5f7) to not make the methods section too long.

21. *It would be interesting to know how many correction notices presented more than one correction and more than one correction type. For example, a correction notice that relates to single typo is quite different to one that has to correct ten typos, and different again to one that corrects ten typos, a hypothesis, and several aspects of the methodology.*

We added information on the number of corrections with more than one type (see footnote 4 on p. 8), and regarding the number of pertinent results per correction (see Figure 4).

22. *Given the relatively small sample size, it would be helpful if percentages were accompanied by the relevant counts (numerators). The relevant denominators could be made a bit clearer too (e.g., “Of the X corrections that involved changes to the results, X (X%) were...”)*

We added counts to supplement the percentages. Doing this also revealed that Table 2 of the original manuscript was based on very small sample sizes. Hence, we decided to remove it, as the observed “pattern” appears rather fragile.

23. *P9 L12 – “we adapted” should this say “we adopted”? “adapted” implies the method was modified but as far as I can tell the same method was used.*

Yes, the same formula was used. We changed it to “adopted”.

24. *The percentage error metric describe around P9 L12 should perhaps be reported in the methods section rather than the results?*

We put the PE metric in the results section because it was not part of the a priori defined coding protocol. It was only after thinking about how to present the results that we thought of using the metric, so to enhance transparency, we would prefer to keep it in the results section.

25. *P15 L12-15 “It is noteworthy that manuscripts submitted to Psychological Science should also conform to these guidelines.” – could it be clarified why? Is PS a member of ICMJE*

We changed this sentence to “It is noteworthy that manuscripts submitted to Psychological Science should “conform to the Recommendations for the Conduct, Reporting, Editing and

Publication of Scholarly Work in Medical Journals, which can be found in full at www.icmje.org (“2020 Submission Guidelines,” n.d.).”

26. P15 L48-49 *“Second, some of the corrections we reviewed still contained errors or even introduced new ones.” This is mentioned in the discussion but I don’t recall seeing any results related to the introduction of new errors in the corrections themselves – apologies if I missed it.*

That’s correct. We did not systematically examine this, but sometimes we happened to notice that a correction contained a mistake or other unusual aspects (e.g., email signature being part of the correction). To make that clear, we rephrased the sentence as follows: “Second, we happened to notice that some corrections still contained errors or even introduced new ones.”

27. P16 L7-9 *“However, we know that some corrections in our sample did not acknowledge the researcher(s) who discovered the error(s).” What is the basis for this statement?*

Personal experience as well as researchers discussing mistakes they found in papers on social media, which later got corrected without any mention of the source. This is now added on p. 17. We could explicitly mention some examples, but this might be misconstrued as a personal “attack”, which is why we decided against it.

28. P16 L17-21 *“In this regard, it is noteworthy that replication studies, another form of scientific self-correction, were very unpopular within psychology (Makel, Plucker, & Hegarty, 2012), but are gradually becoming more mainstream in recent years.” This statement needs empirical support. A more recent estimate is available here: <https://doi.org/10.31222/osf.io/9sz2y>*

Indeed, we added the reference.

29. P17 L48-50 *“A substantial portion of errors in our sample involved misreporting statistical output (e.g., due to a protracted review process).” It’s not clear to me why a protracted review process would lead to misreporting statistical output.*

This was inspired by a particular correction notice. Now, we include a complete quote: “A substantial portion of errors in our sample involved misreporting statistical output, for instance,

"because of errors copying values from previous drafts, missed lines in the SPSS output, and reference to outdated data sets and analyses during a protracted write-up process" (Ramenzoni & Liszkowski, 2017, p. 1)." As an aside, we want to commend these authors for their honesty. We included it to highlight the usefulness of a well-documented data-management plan, and software like R Markdown, not to ridicule them in any way, shape, or form.

30. P17 L10-12 *"In fact, some journals, including Psychological Science, run all to-be-published articles through statcheck by default." I believe they now just ask authors to do this themselves and self-certify the manuscript passed.*

Right, this used to be different. We changed the sentence accordingly.

31. P17 L15-18 *"They could appoint one or more specialists to thoroughly verify the reproducibility of a study's outcomes as a (final) part of the review process." Sakulak et al. (2014) discuss this at length so it might be a citation of interest: <https://doi.org/10.1177%2F1745691614549257>*

Indeed, we added the reference.

Reviewer 2

1. *Bottom Line: I don't disagree with many of the suggestions in the paper. However, I am not sure the content analysis here is necessary to support those suggestions.*

I think the current study needs stronger motivation and many of the analytic decisions need more exposition. Some of the choices may undercut the contribution to the point that it difficult to draw conclusions from the work. The big issues to me where the seemingly ad hoc selection of years and the restriction to just three outlets. In some ways, I think it would be stronger to randomly select corrections from each year and also code journal characteristics by things like policies, field, etc. As it stands, I am not sure what to conclude from what is reported.

Regarding the timeframe: the paper primarily aims to give an idea about the current situation in psychology, it does not aim to provide a complete historic overview. We finished the coding

of corrections in July 2019, which is why the selected time frame ends in 2018 (this is now mentioned in the manuscript on p. 5). The starting point, 2010, is indeed somewhat arbitrary. Going back further would increase sample size, but potentially diminish the relevance regarding the current state of affairs and vice versa. We believe there is not one “correct” starting point, but one has to make a decision. However, we don’t see how this undercuts the contribution.

Regarding the restriction of outlets: a lot of meta-scientific studies in psychology have opted to focus on multiple articles from a limited number of journals (e.g., one in Hardwicke et al. 2018 and Hardwicke et al., 2020; four in Vanpaemel et al., 2015 and Wicherts et al., 2006, two in the Reproducibility Project: Psychology etc.) rather than randomly selecting articles across an entire domain. A random selection of correction notices would have resulted in an amalgam of policies, and quality standards. This is now mentioned on p. 6. Of course, there are disadvantages to our approach, which we acknowledge in the manuscript: “it remains to be seen whether these results generalize to other journals within psychology.” In other words, we don’t think there is one correct way to conduct these kind of studies. Rather, there are several approaches, each with its advantages and disadvantages.

In general, we agree with the reviewer’s bottom line in that part of the discussion does not directly rely on the analysis of the correction notices. Yet, we believe the analysis has merit and is informative in its own right, without detracting from the discussion. In fact, when soliciting feedback on the manuscript before sending it out for formal review, someone explicitly said that “a lot of people have intuitions about what the average correction looks like, but this is the first time I see data.” Our paper does not give a final verdict, but it at least provides a good indication by considering recent correction notices published in well-regarded journals, two of which are also domain-transcending (i.e., *Frontiers in Psychology*, and *Psychological Science*).

2. The paper would benefit from a stronger motivation. What is present in the short introduction hinges on whether there is validity for the proposition that the existence of corrections indicates that science is self-correcting. More from the Grcar (2013) piece could help make this case. To be honest, I don’t think I would argue that the existence of corrections in journals is especially strong evidence one way or the other.

We improved the motivation for the analysis of correction notices, and added some more information about the Grcar (2013) article. Correction notices are recognized as a form of scientific self-correction, not only by Grcar, but also others (e.g., Bordignon, 2020;

Dougherty, 2018). The question we seek to address is whether they yield new insights, and revise theories like other forms of scientific self-correction do, or whether they mostly involve superficial aspects of the scientific record. So, the motivation does not hinge on the validity of the proposition that the existence of corrections indicates that science is self-correcting. It precisely seeks to examine whether corrections indeed reflect a revision of flawed notions and thus whether corrections exemplify that science is self-correcting. We revised the introduction to better reflect this point.

3. The justification for using Scopus needs to be stronger. More details about the initial dataset should be provided including the total number of articles published in each journal and what those time trends looked like from 2010 to 2018. I think that statistic is more informative than just the number of corrections divided by the number of articles published in Psychology as was depicted in Figure 1. Journal specific trends would be more informative.

We used Scopus following Gracar's (2013) approach. This is now made explicit in the manuscript on p. 5. Note that Hubbard (2010) showed that this is a valid method (at least for the chemistry literature). We also added a figure with journal specific trends (i.e., Figure 2), and information regarding the number of articles per journal is provided on p. 5-6.

4. I think stronger justification for examining the years of 2010 to 2018 and for picking just those three journals needs to be in the paper. What was provided might not strike all readers as convincing.

We included additional information/justification for the selection of journals and the timeframe on p. 5-6 (see our response to point 1 of Reviewer 2 for more details).

5. One concern is that different journals/societies may have different norms/expectations about corrections. The authors correctly discuss this. However, I think this has analytic consequences in that corrections are nested within journals with time as a further factor. So, I think a complicated multi-level structure is at play in terms of what is likely to be the data generation mechanism. This issue might be worth considering.

We conducted all analyses for each journal separately, which revealed some interesting outcomes. A further division in terms of year of publication would yield very small sample sizes in many cells, hence we felt it would yield results that are too tricky to interpret.

6. Related to the points about the three outlets, more details about the journals including their policies and how many times editorial teams turned over could be illuminating.

The discussion section describes the policies related to correction notices (insofar that they are available). Note that these policies may have changed at some point during the considered timeframe, but we do not have access to that information as far as we know. Furthermore, as mentioned in the previous point (and now also the manuscript itself), by-journal longitudinal trends are based on small sample sizes, so linking them to potential changes in journal policies or editorial teams would be too speculative in our opinion.

7. The initial agreement on the coding scheme should be reported and discussed. It seemed like the percentage agreement for whether a result is judged to bolster the original conclusion was low (52.63% and I don't think this was corrected for chance agreement). This kind of discussion is important for interpreting the results (i.e., it speaks to the reliability of the coding). The 53% initial figure raised some concerns for me.

The 53% figure already refers to the initial agreement regarding the correction type, perceived implications, and source of the correction, as mentioned in the manuscript. This is indeed a concern, which is why we also look at the quantitative results: the originally reported versus corrected p-values and effect sizes. Note also that part of the reason for this relatively low number is that a) it combines different aspects, so in order for a perfect agreement the correction type, perceived implications and source of correction *all* need to be the same, and b) it considers missing evaluations of one of the coders as a disagreement. Furthermore, the coding scheme is entirely new, because correction notices have not been systematically investigated like this (as far as we are aware), hence it is to be expected that there are some teething problems (e.g., how to deal with corrections issued by the journal: should they be classified as corrections discovered by the journal or not).

8. Some justification for the PE ratio over 10 being classified as a major error should be provided in the report.

Both the PE metric and the error classification criterion were taken from Hardwicke et al., (2018). As far as we know, these were ad hoc decisions without any further justification. We now mention this explicitly: “Hardwicke and colleagues considered PEs greater than 10 to be major numerical errors, though this cutoff is somewhat arbitrary.”